# Invasive Buttonweed *Cotula coronopifolia* (Asteraceae) Is Halotolerant and Has High Potential for Dispersal by Endozoochory

**DOI:** 10.3390/plants13162219

**Published:** 2024-08-10

**Authors:** Raúl Sánchez-García, Andy J. Green, Lina Tomasson, Francisco Hortas, Maria A. Ortiz

**Affiliations:** 1Department of Conservation Biology and Global Change, Estación Biológica de Doñana (EBD), Centro Superior de Investigaciones Científicas (CSIC), Américo Vespucio 26, 41092 Seville, Spain; 2National Coordinator for Aquatic Invasive Alien Species, Swedish Agency for Marine and Water Management, 40439 Gothenburg, Sweden; lina.tomasson@havochvatten.se; 3Department of Biology, Institute of Marine Research (INMAR), University of Cadiz and European University of the Seas (SEA-EU), 11510 Puerto Real, Spain; francisco.hortas@uca.es; 4Department of Vegetal Biology and Ecology, University of Sevilla, Apdo. 1095, 41080 Sevilla, Spain; aortiz@us.es

**Keywords:** endozoochory, dry-fruited seeds, dispersal syndromes, salinity, gut passage, non-indigenous species

## Abstract

Buttonweed (*Cotula coronopifolia*) is native to South Africa but invasive in wetlands in Europe, North America, and Australasia, where it excludes native plants. Despite being dry-fruited, field studies suggest migratory waterbirds can disperse its seeds via gut passage (endozoochory), aiding its expansion. To explore the potential for endozoochory in different regions and habitats, we collected seeds from six populations in Spain, Sweden, and the UK. Germination was tested under different salinity levels (0, 5, 10, 15 g/L) and simulated gut passage treatments: scarification, acidification, or both. No germination occurred at 15 g/L. Higher salinity reduced and delayed germination, but full gut passage treatment (i.e., both scarification and acidification) increased germinability and accelerated germination. Scarification or acid treatment alone resulted in intermediate germination patterns. There were significant salinity × population and gut passage × population interactions on germinability. The acceleration effect of gut passage on germination was stronger at 5–10 g/L than at 0 g/L. This study highlights how migratory birds can facilitate the spread of alien plants introduced by humans. Endozoochory by waterbirds is an understudied mechanism for the long-distance dispersal of dry-fruited alien plants. Further research on *C. coronopifolia*, including population genetics, is necessary to understand dispersal mechanisms and facilitate management strategies.

## 1. Introduction

Biological invasions are one of the most important threats to global biodiversity and a major cause of extinctions [1,2,3]. The number of alien plants in each continent continues to increase, and many of them have major negative impacts on native plant species diversity [1,4]. The invasiveness of alien plants depends partly on many plant traits, on the range of habitats suitable for the species, and on the dispersal mechanisms that allow alien species to expand following their initial introduction [5,6,7]. Dispersal syndromes defined by diaspore morphology have often been used to predict dispersal distances [8], but these syndromes often underestimate the potential for long-distance dispersal [9]. For example, migratory waterbirds disperse many plants by endozoochory (gut passage) which have been attributed to gravity or unassisted syndromes [10,11]. Field studies indicate that waterbird endozoochory is particularly likely for halotolerant plants, which have more avian vectors [12]. They also confirm that waterbirds are frequent vectors of alien plants [13,14].

The buttonweed *Cotula coronopifolia* (Asteraceae; also known as “brass buttons”) is native to South Africa but is a widespread alien species in Europe, North America, and Australia [15]. In Europe, it was first recorded in 1742 in the Netherlands [16], is halotolerant, and is particularly widespread in coastal areas, although it is also common in some inland wetlands [17,18,19,20]. It occupies a wide latitudinal range in Europe of >22°, spanning from southern Spain to Sweden [17]. Rapid recent expansion has occurred along the Swedish Baltic coast, and in the UK and Ireland [19,21]. 

Buttonweed is considered invasive because it excludes native plants (Appendix A), and is reported to promote soil salinization [17,21,22]. Its potential future impact on native communities depends both on its competitive ability across a broad salinity range, and its capacity for dispersal into new habitats. Long distance seed dispersal seems more likely to occur by endozoochory, since *C. coronopifolia* seeds have been repeatedly recorded in waterbird faeces and pellets [23,24,25] and have also been germinated from cattle faeces [21].

Although the species is recorded across a wide salinity range in Europe, there is a lack of previous studies concerning the plasticity of *C. coronopifolia* germination or establishment in response to a salinity gradient, or its geographical variation in halotolerance across the introduced range [20]. Despite previous evidence that seeds can germinate after gut passage, there are no previous studies of the influence of avian ingestion on germinability in this species when compared to control seeds. Van Leeuwen et al. [26] found other wetland plant species to vary widely in their germination patterns in response to simulated avian digestion, with some being favoured (e.g., *Eleocharis palustris*) and others hindered (e.g., *Typha latifolia*). These differences can determine their establishment success after seeds are dispersed into new habitats by waterfowl or other animal vectors [27]. Furthermore, both germination patterns and how they are influenced by gut passage can vary according to the salinity conditions [28,29]. Previous studies on the effects of gut passage on wetland plants have considered only one population per plant species (see [14] for review), but we might expect some populations to be more adapted to endozoochory than others (e.g., due to variation in the density of biotic vectors). 

In this study we tested the germination response of *C. coronopifolia* under a broad salinity range, and how this was influenced by simulated passage through the gut of a waterbird (using a combination of scarification and acid treatments [30]). We compared the response of seeds from several populations representing the broad latitudinal non-native range in Europe. We looked for interactions between population, salinity, and gut passage effects. Given the results for previous studies on native species [28,29,31,32], our initial hypotheses were the following: (i) germination would be inhibited at higher salinities; (ii) populations would differ in their germination responses; (iii) there would be important interactions between salinity and the response to gut passage. We consider the implications of our results for the invasion of habitats of different salinities by *C. coronopifolia* and the potential role of long-distance dispersal via waterbirds in the future expansion of this alien plant.

## 2. Results

### 2.1. Effects of Salinity, Population, and Gut Passage on Germinability

Salinity, gut passage treatment, and population all had highly significant effects on seed germinability (Table 1). Seeds were germinated on Petri dishes prepared with 0, 5 or 10 g/L NaCl solutions (referred to as S0, S5 or S10 from here on). In general, when salinity increased, it had a negative impact on germinability (Figure 1 and Figure 2a). No seeds germinated in S15. 

The final GLM model for germinability also included highly significant interactions between the population and salinity, and between population and gut passage treatment (Table 1). Differences in germinability between populations showed no clear latitudinal trend, and the extremes were represented within Sweden, with SWP having consistently higher germinability than SWH (Figure 2a). Population differences were more pronounced at S10, at which the UK population had significantly lower germinability than all other populations, despite having the second highest germinability at S0 (Figure 2a). Spanish populations (SPC and SPD) had intermediate germinability and exhibited no significant differences. The difference between these Spanish sites in soil salinity (Table 2, Appendix A) was thus not reflected in any differences in germinability (Figure 2a).

Germinability was generally higher for full gut treatment (both acid and scarification) than for control seeds, and partial treatments (acid or scarification) clearly had intermediate germinability at S5 and S10 (Figure 1 and Figure 2). The increase in germinability at full gut treatment compared to controls was particularly strong for SPC, SPD, and SWN (Figure 2b). 

### 2.2. Effects of Gut Passage and Salinity on Time to Germination

Salinity, gut passage treatment, and population all had significant effects on the time to germination for seeds that germinated within 20 days (Table 1). The final model also included a significant interaction between salinity and gut passage treatment (Table 1). 

Overall, the time to germination increased when the salinity increased (Figure 1 and Figure 3a). Seeds subjected to the full gut passage treatment germinated consistently faster than control seeds, and those subjected to partial gut passage had intermediate effects and were significantly different from both controls and full gut treatment at S5 and S10 (Figure 3a). As reflected by the salinity × gut passage interaction, at S0 partial digestion did not have a significant effect on time to germination compared with controls (Figure 3a). There were statistically significant differences in time to germination among populations within Sweden and no evidence of a latitudinal trend across the six populations (Figure 3b). The difference between Spanish sites in soil salinity was not reflected in any significant difference in time to germination (Figure 3b).

## 3. Discussion

We found experimentally that buttonweed *C. coronopifolia* germinates readily across a broad salinity range, responds positively to simulated gut passage (suggesting adaptation or pre-adaptation to endozoochory), and shows broadly similar germination responses from alien populations across a wide latitudinal range. Significant interactions revealed some variation among populations in how germinability was influenced by salinity and by gut passage. Likewise, the effect of gut passage on the time to germination changed significantly with salinity. Our results provide evidence that dispersal by endozoochory plays an important role in the expansion of this invasive plant. 

### 3.1. Effect of Gut Passage on Germination Patterns

The passage of diaspores through avian digestion has a variable effect on the germination of different angiosperm species, including both fleshy- and dry-fruited species [26,33,34]. Even within other Asteraceae found in European wetlands, the response in germinability or germination time to simulated gut passage was positive in some species (e.g., *Eupatorium cannibinum*) but negative in others (e.g., *Tragopogon pratensis*, [26]). Diaspore morphology is diverse, but there is no evidence that seeds from fleshy-fruited angiosperms dispersed by frugivores have an architecture better adapted to resisting gut passage than those of dry-fruited plants dispersed by waterbirds [35]. Our results show an overall increase in germinability and an acceleration in the time to germination for *C. coronopifolia* in response to gut passage. Observed differences with control seeds are likely to have been affected by how we stored our seeds (dry, in the dark, at 4 °C) for months prior to the experiment, which may potentially have increased germination of control seeds by breaking physiological dormancy [29]. If we had stored seeds at room temperature, the effects of gut passage on germination may have been different. Asteraceae (including other species from the genus *Cotula*) are considered to have non-deep physiological dormancy, although the germination responses of each species to fluctuations in light and temperature should be investigated separately [36,37]. Scarification treatments can overcome dormancy in physically dormant seeds. However, when the embryo growth potential is low in physiologically dormant seeds and the pericarp is thick, scarification treatments such as those applied in our simulation may also promote germination by removing the mechanical restraint of the pericarp. This may imply an advantage for seeds dispersed by waterbirds due to the breaking of dormancy during gut passage. In any case, the dormancy strategy of *C. coronopifolia* should be subjected to future research. We would also have been able to interpret our results in more detail if we had tested the viability of seeds before and after the experiment using a tetrazolium test. 

Previously, positive germination responses to gut passage were recorded in non-saline conditions for some other dry-fruited plant species from wet habitats, but many other species showed negative effects [26,29]. We found that this positive effect of gut passage also occurs for *C. coronopifolia* in saline conditions, although there was a salinity × gut passage interaction for time to germination. Halotolerance also increases the number of waterbird species that are likely to disperse a given plant species [12], because migratory waterbirds concentrate in coastal wetlands.

### 3.2. Implications for Long-Distance Dispersal

Seeds of *C. coronopifolia* were previously recovered from waterbird egesta and then germinated [24,25]. Our experiment confirms that the diaspores of this species can resist the passage through the gut of a waterbird, and that gut passage can facilitate germination in saline conditions with a maximum tolerance of ≥10 and <15 g/L NaCl. 

Our full gut passage treatment lasted a total of 4 h, which is comparable to the mean retention times of seeds in the gut of a waterbird [24,38]. However, maximum gut retention times can exceed several days, and so maximum seed dispersal distances by endozoochory can be from several hundred to >1000 km [24,39,40]. It would be interesting to extend the duration of digestion treatments to simulate maximum retention times, and observe their effects on germination. Even outside migratory periods, seeds can be dispersed > 100 km by waterbirds, greatly exceeding expectations for the wind dispersal of *C. coronopifolia* [41]. Alien plant species can gain an advantage over native species by showing higher seed survival and germinability, and lower time to germination, in response to gut passage [27,42].

Our results suggest that germination and establishment are likely to be favoured at salinities of 0–10 g/L after endozoochory events, favouring establishment of new *C. coronopifolia* populations as well as gene flow between existing populations. The relative lack of differences in germination response we found among populations is consistent with latitudinal connectivity and gene flow over the recent decades and centuries since this alien species colonized Europe. This would be expected if there is a major role played by waterbird dispersal vectors. Population effects were generally much weaker than the effects of salinity and gut passage, and we found no evidence of local adaptation when comparing nearby populations living in different soil salinities. 

### 3.3. The Importance of Halotolerance

Germination of *C. coronopifolia* remained high at 10 g/L NaCL in all but one population (UK). This exceptional population may have a unique origin, since there was a direct introduction reported into nearby gardens by the year 1886 [43]. The halotolerance we observed in all other populations exceeds that of many native mudflat species in Europe [44,45] and helps explain how this alien species has occupied both freshwater and brackish habitats, including coastal marshes and wetlands in the Mediterranean climatic zone that vary in salinity over the year (see [46] for SPD). Hydrological changes due to climate change and water extraction for irrigation and other human activities are causing wetland salinization in many areas [47], and this halotolerance may allow *C. coronopifolia* to colonise further habitats in the future and to outcompete native species less adapted to salt stress. Invasion of this plant may itself promote further salinization [17]. The observed expansion in the species’ distribution over recent decades in Europe is consistent with these expectations and with an important role for dispersal being played by waterbird vectors (e.g., to isolated, inland wetlands, [48]). 

## 4. Materials and Methods

### 4.1. Study Species

Buttonweed *C. coronopifolia* (Asteraceae) is a decumbent and stoloniferous herb that can spread vegetatively by growing roots on the stem nodes. It is an annual or short-lived perennial plant [18,21] which inhabits wetlands, salt and freshwater marshes, and temporary pools. It is considered a pioneer species with an ability to self-fertilise, high seed production, and efficient seed dispersal [15]. It can be considered a halotolerant mudflat species (Ellenberg F = 7, Ellenberg S = 3, [49]).

The plant has heads with yellow disk flowers, the outer ones being female and the others hermaphroditic, with plano-convex achenes (cypsela) with marginal thickening [50]. Van Der Toorn et al. (1980) estimated seed production at 20,000–50,000 diaspores per individual plant, and considered cross-pollination to occur by wind and insects and seeds to float for <10 min. However, more recently, floating time was estimated at 29–40 h (L. Tomasson, unpublished data). Like many Asteraceae, it has been assigned an anemochory dispersal syndrome [49], but it is debatable whether the achenes have traits that are adaptations for wind dispersal (Appendix A). Although *C. coronopifolia* may perhaps have initially reached Europe from South Africa via boat traffic, its spread may also be partly driven by its use as an ornamental plant, dating back at least to the 19th century [43]. It is currently traded legally in Europe as a pond plant. 

### 4.2. Study Sites and Plant Material

We collected mature flower heads from *C. coronopifolia* individuals with mature achenes containing the seeds at six different sampling sites (Table 1). Heads from about 20 individuals were collected from each population, in three different countries: Spain, Sweden, and the United Kingdom. Four populations were essentially coastal, including the Atlantic coast in Spain (SPC in Table 1), the Irish Sea in the UK (UK), and the Baltic Sea and Kattegat in Sweden (SWN and SWH). SPD and SWP were populations from inland wetlands with low salinities. The sites sampled in Sweden have a low salinity compared to those near the Atlantic and Irish Sea. The area adjacent to SWN has a lower and less fluctuating salinity than SWH, which is located at Kattegat. The area around SWH has a variable surface salinity due to the outlet between the Baltic and North Seas. Salinities for Swedish sites for Table 2 were obtained from [51]. Soil salinities for Spanish sites were obtained from six soil sampling points per location. The sample from each point was sieved to less than 2 mm in size and three subsamples of 15 g each were measured using a precision scale. The subsamples were dried for 48 h at 40 °C in an oven and then mixed in a 1:1 proportion with 30 mL of distilled water before being centrifuged for 15 min at 4000 rpm. The electrical conductivity (Ec) (mS cm^−1^), was then measured at 25 °C in the supernatant using a Portable Crison CM35 electrical conductivity metre (Crison Instruments S.A., Sevilla, Spain). The Ec of each soil sample and each point was calculated, respectively, as the average of the three subsamples and as the average of six corresponding samples per location.

The plant samples were collected from April to August in 2022 and stored in dry conditions, in paper bags in darkness at 4 °C until the experiment started in February 2023. As far as we know, no one has formally classified the dormancy strategy of this species, but other Asteraceae exhibit non-deep physiological dormancy [36]. Seeds were separated from each of the six populations, with an even representation of available flower heads (Table 2).

### 4.3. Gut Passage Simulation

To test how passage through the gut of a waterbird modifies seed germination patterns at different salinities, we performed a gut passage simulation, following the methodology of [30,52]. Seeds of each population were randomly divided into four groups (40 seeds per group), including a control group. The other groups were subjected to three digestion treatments: seed scarification simulating grit in the waterbird gizzard, chemical treatment with hydrochloric acid, or a combination of both (scarification followed by acid). For the scarification step, we added the seeds to a 150 mL plastic flask with 10 g of grit (2–4 mm size) and 4 mL of water. The flasks were then placed in an incubator (New Brunswick Classic^®^ C24 Incubator Shaker, Hampton, NH, USA) for 2 hr at 42 °C and 300 shakes/min. Before continuing with the chemical step, we separated the seeds from the grit using a stack of two sieves, one with a 2 mm mesh to retain grit and another with a 300 µm mesh to retain seeds. Then, seeds were immersed in 5 mL of a HCl solution at pH = 2.5 for 2 h at 42 °C without shaking. After this second step, we retrieved the seeds and rinsed them with distilled water.

### 4.4. Germination Experiment

Seeds from gut treatments and controls (Appendix A) were placed on 90 × 15 mm Petri dishes (with four compartments) with agar (0.8%) mixed with 0, 5, 10, or 15 g/L NaCl solutions. A total of 96 Petri dishes were used as the resulting combination of four salinities, four treatments, and six populations, with 25–35 seeds per dish. Two populations were sown in each dish, in different compartments, and each population was divided between two separate dishes.

The Petri dishes with seeds were placed in a germination chamber (25/20 °C, 70–80% relative humidity, and a 16/8 h light/dark period) for 20 days (6 to 26 February 2023), considered a long enough period due to the generally rapid germination of plants in the family Asteraceae. Germination data were recorded daily from day 3 onwards, counting as germinated those that were showing a hypocotyl of 4–6 mm and visible cotyledons. 

### 4.5. Statistical Analyses

No seeds germinated in the highest saline treatment (NaCl = 15 g/L), so this treatment was excluded from statistical analyses. The influence of the different treatments on the gut passage simulation (factor of four levels), salinity (factor of three levels) and population (factor of six levels) on the germination pattern was modelled with generalised linear models (GLMs). We analysed the germination response (Yes = 1, or No = 0) with a binomial error distribution and logit function. We analysed the time to germination (i.e., days taken to germinate) with a negative binomial error distribution and log-link function, excluding seeds that failed to germinate. This error distribution was chosen because a Poisson distribution was found to be inappropriate owing to overdispersion.

Global models with all covariates (salinity, population, and gut passage simulation) and their first order interactions were used for both dependent variables, germinability and time to germination. Final model selection was based on the lowest value of the corrected Akaike information criteria [AICc]. Statistical significance and contribution to variability were tested for the fixed terms and their interactions. Differences among the factor levels in the fixed terms were further tested using False Discovery Rate (FDR) post hoc tests. Generalised linear model analyses were performed and checked using the packages DHARMa and lmer under R version 4.3.2 for Windows (R Core Team 2015). 

## 5. Conclusions

Our experimental study complements previous field studies and provides strong evidence for the importance of avian endozoochory in the invasion of a dry-fruited Asteraceae. In contrast to the extensive literature on dispersal of native plants by waterbirds [14], our study represents one of the most detailed examples to date of how waterbird endozoochory can spread alien plants (see [13,41,53] for other examples). 

To understand the connectivity between European populations of this alien species—and the relative importance of alternative dispersal mechanisms (including hydrochory and human vectors)—genetic studies are required that also relate population differences to waterbird movement patterns. The potential for hydrochory of *C. coronopifolia* should be assessed in detail with buoyancy experiments (e.g., [53]). These studies would help to evaluate the importance of long-distance endozoochory for *C. coronopifolia* compared to other dispersal mechanisms such as wind, water, or anthropic activity.

## Figures and Tables

**Figure 1 plants-13-02219-f001:**
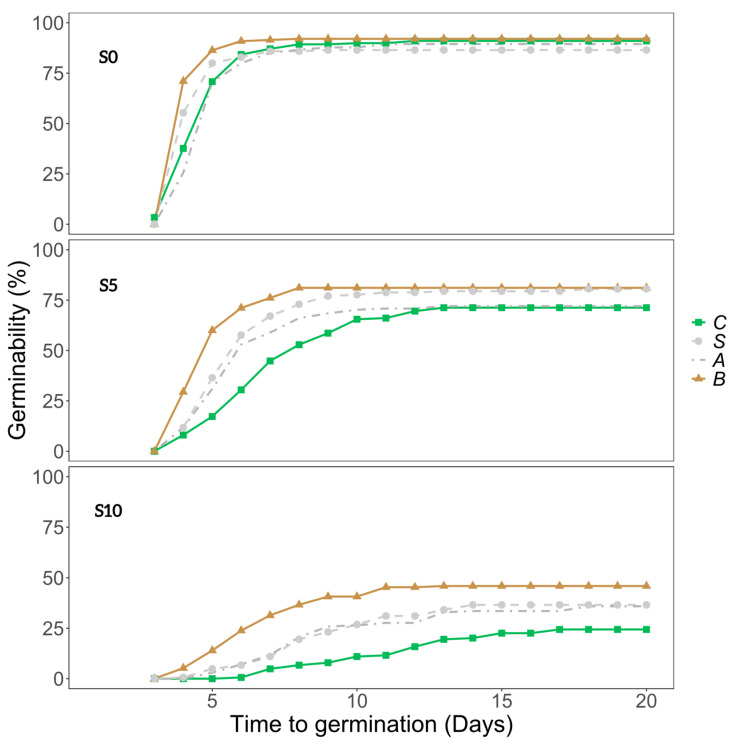
Cumulative germination plots of *Cotula coronopifolia* seeds from all the populations in NaCl = 0 g/L (S0), NaCl = 5 g/L (S5), and NaCl = 10 g/L (S10) for the different treatments in the gut passage simulation: control (C), scarification (S), acid (A), or both (B).

**Figure 2 plants-13-02219-f002:**
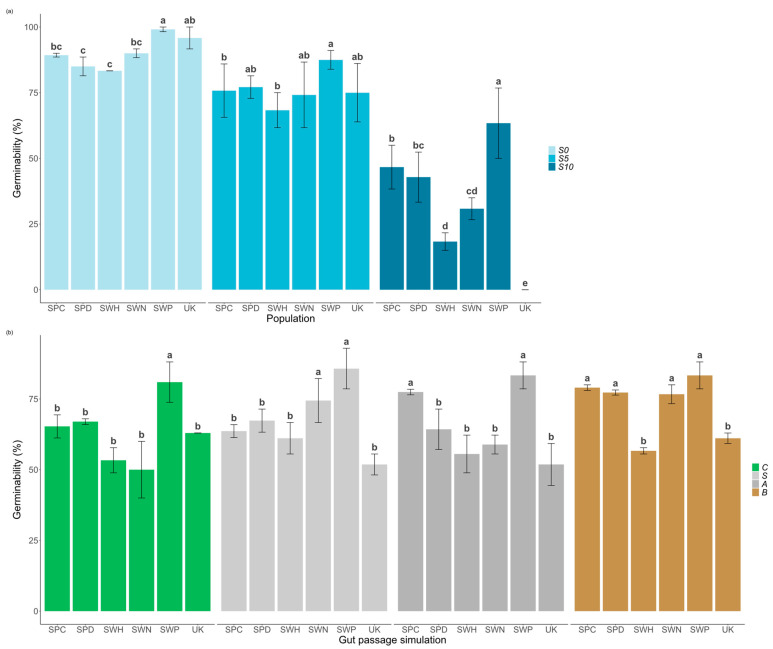
Germinability of *Cotula coronopifolia* seeds from each population by salinity (**a**): NaCl = 0 g/L (S0), NaCl = 5 g/L (S5), and NaCl = 10 g/L (S10), and by gut passage treatment (**b**): control (C), scarification (S), acid (A), or both (B). The bars represent means ± Standard Error (SE). Bars with different letters within populations (comparisons within a given (**a**) salinity or (**b**) gut passage treatment) differ significantly (*p* < 0.05) in post hoc tests from the germinability GLM of Table 1.

**Figure 3 plants-13-02219-f003:**
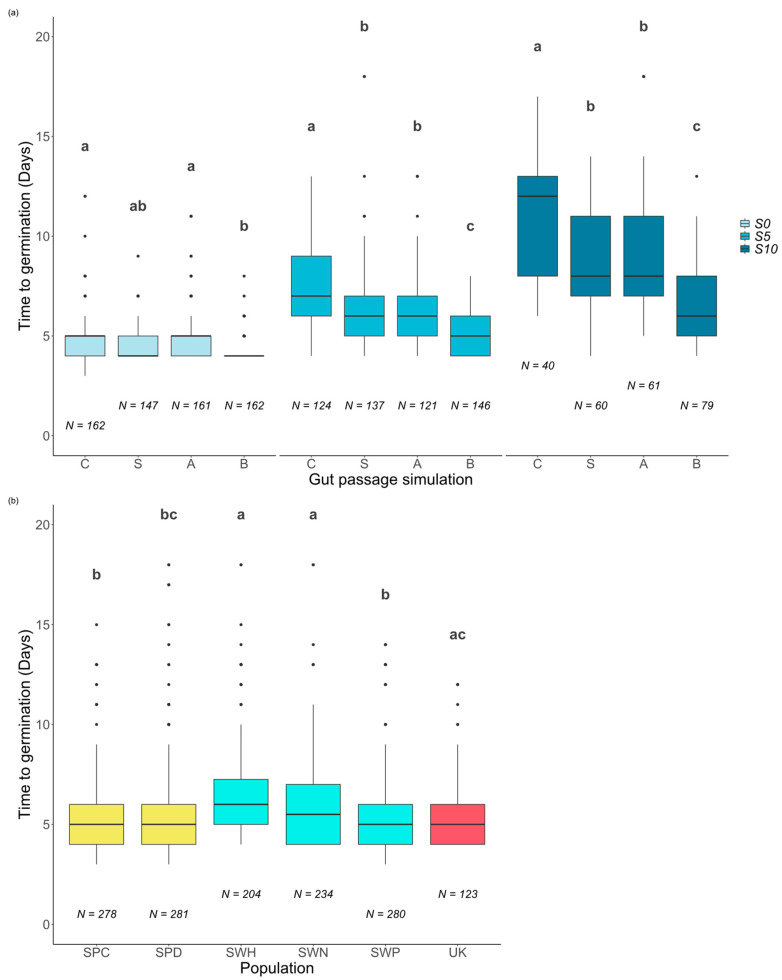
Time to germination for *Cotula coronopifolia* seeds under different gut passage treatments: (**a**) control (C), scarification (S), acid (A), or both (B) by salinity: NaCl = 0 g/L (S0), NaCl = 5 g/L (S5), and NaCl = 10 g/L (S10). Time to germination for seeds by population (**b**). The line within each box represents the median (Q2). Q1 and Q3 are represented by the lower and the upper edges. Lines extending from the top and the bottom of the box represent the variability outside the quartiles, while individual data points are outliers. Boxes with different letters are as follows: (**a**) gut passage treatment (comparisons within a given salinity), and (**b**) populations differ significantly (*p* < 0.05) in post hoc tests for the GLM from Table 1.

**Table 1 plants-13-02219-t001:** Final GLM models for germinability and time to germination, showing details of the fixed term effects and their interactions.

Germinability	Df	Sum Sq	Mean Sq	F Value	Pr (>|F|)
Salinity	2	107.51	53.75	355.326	<0.001
Population	5	15.59	3.12	20.615	<0.001
Gut passage	3	3.07	1.02	6.770	<0.001
Salinity × Population	10	12.43	1.24	8.215	<0.001
Gut passage × Population	15	5.61	0.37	2.470	0.00133
**Time to Germination**					
Salinity	2	2686	1343.1	454.56	<0.001
Population	5	298	59.7	20.20	<0.001
Gut passage	3	604	201.4	68.17	<0.001
Salinity × Gut passage	6	330	54.9	18.60	<0.001

**Table 2 plants-13-02219-t002:** Details of the sampling sites where *Cotula coronopifolia* seeds were collected.

Sampling Site	Locality	Latitude	Longitude	Sampling Date	Conductivity/ Salinity
SPC	Salinas de Cetina, Cádiz, Spain	36°34′31″ N	6°08′32″ W	8 April 2022	4.66 mS/cm ^1^
SPD	Laguna Dulce, Doñana, Spain	36°58′51.95″ N	6°29′5.53″ W	8 April 2022	0.79 mS/cm ^1^
SWH	Halland. Sweden	57°01′20.3″ N	12°19′53.4″ E	21 August 2022	18–26 ppt ^2^
SWN	Nyköping. Sweden	58°42′53.7″ N	17°05′15.1″ E	12 August 2022	6–7 ppt ^2^
SWP	Pulken. Sweden	55°52′59.8″ N	14°12′21.6″ E	5 August 2022	<1 ppt ^2^
UK	Hoylake, Wirral. United Kingdom	53°23′36.1″ N	3°11′09.7″ W	29 July 2022	-

^1^ For SPC and SPD, soil conductivity is presented (mean of three replicates from each of six sampling points). ^2^ For SWH, SWN and SWP, salinity of the water surface is reported.

## Data Availability

The data are presented in this paper as a Appendix A.

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
