# Peer review of "Invasive Buttonweed *Cotula coronopifolia* (Asteraceae) Is Halotolerant and Has High Potential for Dispersal by Endozoochory"

_plants, 2024, doi:10.3390/plants13162219_

Round 1
Reviewer 1 Report
Comments and Suggestions for Authors
Line 51. Change “22o” to “22°”
Line 57. Change “as” to “since”
Line 58. Delete comma before “and” You do not have a complete clause after the “and”, thus a comma is not correct.
Line 64. Starting a sentence with “[25]” is very awkward. I suggest you provide the name of the first author and then have “[25}”
Line 83. Delete comma before “and”
Line 87. Change “petri” to “Petri” This is a person’s name.
Line 132. Delete comma before “and”
Lines 163-165. This part is very confusing. How did you store the seeds? Were the seeds on a moist substrate during storage? If seeds were not on a moist substrate during cold storage, then they did not receive a cold stratification treatment. However, dormancy break can (and often does occur in Asteraceae) when seeds are in dry cold storage. Dormancy-break during dry storage (regardless of the temperature) is call afterripening; however, (as you say) afterripening is faster at high than at low temperatures. Need to adjust these lines to match the storage conditions.
Line 198. Change ”as” to “since”
Line 208. Delete comma before “and”
Line 213. Change “which” to “that”
Line 218. Change “presents” to “has”
Line 219. Change “hermaphrodite” to “hermaphroditic”
Line 221. Individual what? Do you mean “individual plant” ? Not clear
Line 221. Delete comma before “and”
Line 284. Change “was” to “were”
References – need to check the style. In a few references you have an upper case letter for all the words, but in others you do not – need to be consistent and follow style of Plants.
Line 328-329; 363-363; 411-412
Lines. 372, 382. Put species name in italics.
Comments on the Quality of English LanguageOnly minor changes are needed - see comments to authors.
Author Response
Thank you very much for taking the time to review this manuscript. Please find the detailed responses below:
Comments 1: Line 51. Change “22o” to “22°”.
Response 1: Thank you for pointing this out. We have changed the text in Line 51. "... in Europe of >22˚ spanning ...".
Comments 2: Line 57. Change “as” to “since”. Line 198. Change ”as” to “since”.
Response 2: We agree with this comment. Therefore we have changed the text in Line 58: "... occur by endozoochory, since C. coronopifolia seeds ..." and in Line 209: "... have a unique origin, since there was ...".
Comments 3: Line 58. Delete comma before “and” You do not have a complete clause after the “and”, thus a comma is not correct. Line 83. Delete comma before “and”. Line 132. Delete comma before “and”. Line 208. Delete comma before “and”. Line 221. Delete comma before “and”.
Response 3: Agree. We have deleted the comma before and in the text in Lines 59, 84, 136, 219 and 233.
Comments 4: Line 64. Starting a sentence with “[25]” is very awkward. I suggest you provide the name of the first author and then have “[25}”.
Response 4: We agree with this suggestion, so we have added the name of the first author in the text in Line 59. "Van Leeuwen et al. [26] found other ...".
Comments 5: Line 87. Change “petri” to “Petri” This is a person’s name.
Response 5: We have added the capital letter in the word Petri in the text in Line 89.
Comments 6: Lines 163-165. This part is very confusing. How did you store the seeds? Were the seeds on a moist substrate during storage? If seeds were not on a moist substrate during cold storage, then they did not receive a cold stratification treatment. However, dormancy break can (and often does occur in Asteraceae) when seeds are in dry cold storage. Dormancy-break during dry storage (regardless of the temperature) is call afterripening; however, (as you say) afterripening is faster at high than at low temperatures. Need to adjust these lines to match the storage conditions.
Response 6: Thank you for pointing this out. We agree with this comment. Therefore we have changed the text in Lines 252-253. "The samples were collected from April to August in 2022 and stored in dry conditions, in paper bags in darkness at 4 °C until the experiment started in February 2023."
Comments 7: Line 213. Change “which” to “that”.
Response 7: We applied the changes in Lines 224-225. "... stoloniferous herb that can spread ...".
Comments 8: Line 218. Change “presents” to “has”.
Response 8: Change done in Line 230. "The plant has heads with ...".
Comments 9: Line 219. Change “hermaphrodite” to “hermaphroditic”.
Response 9: We have changed the text in Line 231. "others hermaphroditic, ...".
Comments 10: Line 221. Individual what? Do you mean “individual plant” ? Not clear.
Response 10: For clarity, we changed the text in Lines 232-233. "... estimated seed production at 20,000-50,000 diaspores per individual plant, considered cross-pollination ...".
Comments 11: Line 284. Change “was” to “were”.
Response 11: Change done in the text in Lines 297. "... variability were tested ...".
Comments 12: References – need to check the style. In a few references you have an upper case letter for all the words, but in others you do not – need to be consistent and follow style of Plants.
Line 328-329; 363-363; 411-412.
Response 12: We have revised the references format and applied the Plants style in the text in Lines 340-341, 373-374.
Comments 13: Lines. 372, 382. Put species name in italics.
Response 13: We have revised the species name and applied the changes in the text in Lines 386, 402.
Reviewer 2 Report
Comments and Suggestions for Authors
1. line 51 : >22o --> What does this notation mean? Unit?
2. line 64: '[25]' found other ~~ --> The sentence that uses the document number as the subject is awkward. It would be better to modify it to be more natural.
3. line 71 : 's (see Green et al. 2023 for review),' --> The citation format is awkward. It needs to be modified to match the format of this journal, such as the reference number.
4. line 87 : petri dishes --> In Petri, p must be capitalized.
5. Figure 2: The significance characters in the graph are strange overall. The notation of significance characters according to the numbers is inconsistent. There are many parts that do not match the interpretation of the results. It needs to be checked again.
6. line 106 ~ 111 : The interpretation is too general. Is it based on statistical significance? The entire interpretation needs to be revised.
7. Figure 3: Statistical significance characters in the graph, the processing results are questionable. It seems that re-examination and interpretation are necessary.
Author Response
Thank you very much for taking the time to review this manuscript. Please find the detailed responses below:
Comments 1: line 51 : >22o --> What does this notation mean? Unit?
Response 1: Thank you for pointing this out. We have changed the text in Line 51. "... in Europe of >22˚ spanning ...".
Comments 2: line 64: '[25]' found other ~~ --> The sentence that uses the document number as the subject is awkward. It would be better to modify it to be more natural.
Response 2: We agree with this suggestion, so we have added the name of the first author in the text in Line 59. "Van Leeuwen et al. [26] found other ...".
Comments 3: line 71 : 's (see Green et al. 2023 for review),' --> The citation format is awkward. It needs to be modified to match the format of this journal, such as the reference number.
Response 3: We have added that citation as a reference in Line 73.
Comments 4: line 87 : petri dishes --> In Petri, p must be capitalized.
Response 4: We have added the capital letter in the word Petri in the text in Line 89.
Comments 5: Figure 2: The significance characters in the graph are strange overall. The notation of significance characters according to the numbers is inconsistent. There are many parts that do not match the interpretation of the results. It needs to be checked again.
Response 5: Agree. We have modified the significance letters according to the post hoc results. The original graphs contained errors in the assignment of significance letters.
Comments 6: line 106 ~ 111 : The interpretation is too general. Is it based on statistical significance? The entire interpretation needs to be revised.
Response 6: We decided to focus the interpretation of the results on the results of statistical significance, but to avoid over-interpretation due to the limitations of the experiment and the technique used.
Comments 7: Figure 3: Statistical significance characters in the graph, the processing results are questionable. It seems that re-examination and interpretation are necessary.
Response 7: Agree. We have modified the significance letters according to the post hoc results as in Figure 2. Now, both Figure 1 and 2 are modified and both have an additional sentence clarifying the significance. "Figure 2. Germinability of C. coronopifolia seeds from each population by salinity (a): NaCl = 0 g/L (S0), NaCl = 5 g/L (S5) and NaCl = 10 g/L (S10) and by gut passage treatment (b): control (C), scar-ification (S), acid (A) or both (B). The bars represent means ± Standard Error (SE). Bars with different letters within populations (comparing only in the same (a) salinity or (b) gut passage treatment) differ significantly with P < 0.05 in post hoc tests from germinability GLM." and "Figure 3. Time to germination for C. coronopifolia seeds under different gut passage treatments (a): control (C), scarification (S), acid (A) or both (B) by salinity: NaCl = 0 g/L (S0), NaCl = 5 g/L (S5) and NaCl = 10 g/L (S10). Time to germination for seeds by population (b). The line within the box represents the median (Q2). Q1 and Q3 are represented by the lower and the upper edge. Lines extending from the top and the bottom represents the variability outside the quartiles, while in-dividual data points are outliers. Boxes with different letters within (a) gut passage treatment (comparing only in the same salinity) and (b) populations differ significantly with P < 0.05 in post hoc tests from time to germination GLM.".
Reviewer 3 Report
Comments and Suggestions for Authors
This research investigated the effect of simulated passage through the gut of a waterbird, and salinity, on germination of an invasive buttonweed – Cotula coronopifolia (Asteraceae). There are no previous studies on the influence of avian ingestion on germinability of this species. Seeds were collected from 6 populations in Spain, Sweeden and the UK. Gut passage was simulated using scarification or acidification treatments, or both. Three different salinity levels were investigated. Both ingestion treatments applied together (defined as ‘full gut passage’) increased and accelerated germination, particularly in saltwater compared to fresh.
This is a thorough, rigorous and well written study. Results do support the notion that dispersal by waterbirds plays a role in the expansion of this invasive plant.
Abstract –
Suggest authors define ‘full gut passage’ as both the scarification and acidification treatments applied together.
‘non-classical endozoochory’ – I think this refers to dispersal of invasive plants, but this could be better explained/defined throughout since it is an important point of difference.
Introduction -
The potential for endozoochory was established prior to this study - viable C. coronopifolia seeds have been repeatedly recorded in waterbird faeces and pellets. Therefore, to avoid confusion and highlight the novelty of this study and the literature gap it addresses, I suggest shifting the focus of the Introduction (and paper title) away from the question of whether seeds of this species are dispersed by birds, and towards the effect of avian digestion on germination under a range of salinity.
I also suggest that some basic ‘simulated avian digestion’ methods/definitions are included in the Intro, and referenced, for clarity.
Results -
Consider putting the sentence ‘Salinity, gut passage treatment and population all had highly significant effects on seed germinability (Table 1)’ at the beginning of Section 2.1, rather than part way through.
Figure legends – add species name.
Discussion
Line 179 – maximum gut retention times suggest that longer durations of ingestion treatments need investigating.
Line 292 – ‘one of the best examples to date’ – explain what is meant by this.
Author Response
Thank you very much for taking the time to review this manuscript. Please find the detailed responses below:
Comments 1: Suggest authors define ‘full gut passage’ as both the scarification and acidification treatments applied together.
Response 1: We agree with this comment, we have changed the text in Line 21. "... full gut passage treatment (i.e. both scarification and acidification) ...".
Comments 2: ‘non-classical endozoochory’ – I think this refers to dispersal of invasive plants, but this could be better explained/defined throughout since it is an important point of difference.
Response 2: We deleted "non-classical" to avoid confusion.
Comments 3: The potential for endozoochory was established prior to this study - viable C. coronopifolia seeds have been repeatedly recorded in waterbird faeces and pellets. Therefore, to avoid confusion and highlight the novelty of this study and the literature gap it addresses, I suggest shifting the focus of the Introduction (and paper title) away from the question of whether seeds of this species are dispersed by birds, and towards the effect of avian digestion on germination under a range of salinity. I also suggest that some basic ‘simulated avian digestion’ methods/definitions are included in the Intro, and referenced, for clarity.
Response 3: We made some changes to deal with these comments, adding explanation for the methods, and clarifying that we are not studying whether seeds germinate after gut passage or not. Rather, we are studying the difference between gut passage and control seeds, and this is a novel and important part of our study. For this reason, we do not think it is necessary to modify the title.
Comments 4: Consider putting the sentence ‘Salinity, gut passage treatment and population all had highly significant effects on seed germinability (Table 1)’ at the beginning of Section 2.1, rather than part way through.
Response 4: Agree. We have changed the sentence in Line 90-91 as you suggested. "Salinity, gut passage treatment, and population all had highly significant effects on seed germinability (Table 1).".
Comments 5: Figure legends – add species name.
Response 5: We have added the species name in the 3 figures.
Comments 6: Line 179 – maximum gut retention times suggest that longer durations of ingestion treatments need investigating.
Response 6: Agree. We have added a new sentence in this paragraph to highlight that suggestion. Line 195-196. "It would be interesting to extend the duration of digestion treatments to simulate maxi-mum retention times, and their effects on germination.".
Comments 7: Line 292 – ‘one of the best examples to date’ – explain what is meant by this.
Response 7: We reworded the sentence to improve clarity. Lines 308-311. ". In contrast to the extensive literature on dispersal of native plants by waterbirds [4814], our study represents one of the most detailed examples to date of how waterbird endozoochory can spread alien plants (see [13,40,51] for other examples). "
Reviewer 4 Report
Comments and Suggestions for Authors
The authors expose a study of the germinability and the time to germination of an invasive plant species. They compare different species sources and resistance to salinity to understand the halotolerance and they test treatments (mechanical scarification and acidic treatment in moderate temperature conditions) simulating gut transit to test the hypothesis of adaptation or pre-adaptation of the species to endozoochory. The hypothesis they want to test is that water birds shallowing the seeds are potential vectors of long-range dispersion.
The study and the redaction are carefully carried out and I have a limited number of minor questions or suggestions. Yet, it is lacking. For me, the experimental work or the literature analysis should bring the dormancy class of the species (see Baskin, J.M. and Baskin, C.C., A classification system for seed dormancy, Seed Sci. Res., 2004, vol. 14, pp. 1–16. https://doi.org/10.1079/SSR2003150, Baskin, C.C. and Baskin, J.M., The natural history of soil seed banks of arable land, Weed Sci., 2006, vol. 54, pp. 549–557. https://doi.org/10.1614/WS-05-034R.1, Baskin, C.C. and Baskin, J.M., Seeds: Ecology, Biogeography, and, Evolution of Dormancy and Germination, Amsterdam: Elsevier, 2014. Baskin, C. and Baskin, J., Plant Regeneration from Seeds: A Global Warming Perspective, Cambridge: Cambridge Univ. Press, 2022) since it would fix a theoretical framework. In addition, the authors suppose (L160-166) that their cold storage may have broken physiological dormancy but they do not prior establish this property (see for instance Long, R.L., Gorecki, M.J., Renton, M., Scott, J.K., Colville, L., Goggin, D.E., Commander, L.E., Westcott, D.A., Cherry, H., and Finch-Savage, W.E., The ecophysiology of seed persistence: A mechanistic view of the journey to germination or demise, Biol. Rev., 2015, vol. 90, pp. 31–59. https://doi.org/10.1111/brv.12095). Another problem with the experimental work is that the initial and the final viability of the seeds of the different populations is not determine.
Author Response
Thank you very much for taking the time to review this manuscript. Please find the detailed responses below:
Comments 1: For me, the experimental work or the literature analysis should bring the dormancy class of the species (see Baskin, J.M. and Baskin, C.C., A classification system for seed dormancy, Seed Sci. Res., 2004, vol. 14, pp. 1–16. https://doi.org/10.1079/SSR2003150, Baskin, C.C. and Baskin, J.M., The natural history of soil seed banks of arable land, Weed Sci., 2006, vol. 54, pp. 549–557. https://doi.org/10.1614/WS-05-034R.1, Baskin, C.C. and Baskin, J.M., Seeds: Ecology, Biogeography, and, Evolution of Dormancy and Germination, Amsterdam: Elsevier, 2014. Baskin, C. and Baskin, J., Plant Regeneration from Seeds: A Global Warming Perspective, Cambridge: Cambridge Univ. Press, 2022) since it would fix a theoretical framework. In addition, the authors suppose (L160-166) that their cold storage may have broken physiological dormancy but they do not prior establish this property (see for instance Long, R.L., Gorecki, M.J., Renton, M., Scott, J.K., Colville, L., Goggin, D.E., Commander, L.E., Westcott, D.A., Cherry, H., and Finch-Savage, W.E., The ecophysiology of seed persistence: A mechanistic view of the journey to germination or demise, Biol. Rev., 2015, vol. 90, pp. 31–59. https://doi.org/10.1111/brv.12095).
Response 1: Thank you. As far as we know, no one has formally classified the dormancy strategy of this species, but the classifications of Baskin & Baskin for other Asteraceae suggest it would be non-deep physiological dormancy. We have now reworded the text and added an additional reference to address this question. Lines 168-175. "Observed differences with control seeds are likely to be affected by how we stored our seeds (dry, in the dark, at 4 ºC) for months prior to the experiment, which may have increased germination of control seeds by breaking physiological dormancy [29]. If we had stored seeds at room temperature, the positive effects of gut passage on germination may potentially have been even stronger. Asteraceae are considered to have non-deep physiological dormancy, but the germination responses of each species to fluctuations in light and temperature should be investigated separately [36].". Lines 258-260. "As far as we know, no one has formally classified the dormancy strategy of this species, but other Asteraceae exhibit non-deep physiological dormancy [36]. "
Comments 2: Another problem with the experimental work is that the initial and the final viability of the seeds of the different populations is not determine.
Response 2: It would have been interesting to look at the loss over time in total viability for different populations using a tetrazolium test, but this was outside the scope of our current study.
Round 2
Reviewer 2 Report
Comments and Suggestions for Authors
Many parts have been revised and supplemented.
However, there is still a lack of consistency in the statistical processing results.
Depending on the post hoc test, there is a lack of uniformity in the notation of whether the result value starts with a for a large number or a for a small number.
This lacks the logic for consistently interpreting the results.
Author Response
Comments:
Many parts have been revised and supplemented.
However, there is still a lack of consistency in the statistical processing results.
Depending on the post hoc test, there is a lack of uniformity in the notation of whether the result value starts with a for a large number or a for a small number.
This lacks the logic for consistently interpreting the results.
Response:
We now understand precisely what the Reviewer means, and have now changed the notation as requested, to ensure that “a” always represents the highest value, and “e” (or the other last letter in the series) always represents the lowest value. This is the logic that the reviewer is requesting, which was missing from previous versions.
Reviewer 4 Report
Comments and Suggestions for Authors
The authors do not respond to my comments. My opinionis that they have to determine the dormancy class of their species but they report that the Asteraceae have generally non-deep physiological dormancy. There are 32,000 Asteraceae.
The other point is the determination of the viability of the seed samples to be able to discuss the results correctly. Again, they reply is not satisfying : it would be interesting but out of our scope
Author Response
Comment 1:
The authors do not respond to my comments.
Response 1:
We are sorry that the reviewer feels this way, as we did provide a response. Perhaps something was not explained adequately. We have now provided further changes to the text, and also respond below.
Comment 2:
My opinionis that they have to determine the dormancy class of their species but they report that the Asteraceae have generally non-deep physiological dormancy. There are 32,000 Asteraceae.
Response 2:
We have now added text reading “Asteraceae are considered to have non-deep physiological dormancy, but the germination responses of each species to fluctuations in light and temperature should be investigated separately [36]. Seeds of other species from the genus Cotula also exhibit non-deep physiological dormancy, which can be overcome through scarification treatments such as those applied in our simulation. This may imply an advantage for seeds dispersed by waterbirds due to the breaking of dormancy during gut passage [37]. The dormancy strategy of C. coronopifolia should be subjected to future research. ”. This has not been done as yet, and requires a complicated and independent study that cannot possibly be included in our revision.
Comment 3:
The other point is the determination of the viability of the seed samples to be able to discuss the results correctly. Again, they reply is not satisfying : it would be interesting but out of our scope
Response 3:
We have now added text reading “We would also have been able to interpret our results in more detail if we had tested the viability of seeds before and after the experiment using a tetrazolium test”. Hence we have made it clear that this would have been useful, but it is not possible after the event. It would require us to repeat the whole experiment, and this cannot possibly be included in our revision.
Round 3
Reviewer 2 Report
Comments and Suggestions for Authors
it was well revised
Author Response
Comments 1: it was well revised
Response1: I have to response in order to upload the new version.